

# Influence of irrigation on root zone storage capacity estimation

Fransje van Oorschot[1,2], Ruud van der Ent[1], Andrea Alessandri[2], and Markus Hrachowitz[1]

[1]Department of Water Management, Faculty of Civil Engineering and Geosciences, Delft University of Technology, Delft, The Netherlands

[2]Institute of Atmospheric Sciences and Climate, National Research Council of Italy (CNR-ISAC), Bologna, Italy

**Correspondence:** Fransje van Oorschot (f.vanoorschot@tudelft.nl)

**Abstract.** Vegetation plays a crucial role in regulating the water cycle through transpiration, which is the water flux from the subsurface to the atmosphere via vegetation roots. The amount and timing of transpiration is controlled by the interplay of seasonal energy and water supply. The latter strongly depends on the size of the root zone storage capacity ($S_r$) which represents the maximum accessible volume of water that vegetation can use for transpiration. $S_r$ is primarily influenced by hydro-climatic conditions as vegetation optimizes its root system in a way it can guarantee water uptake and overcome dry periods. $S_r$ estimates are commonly derived from root zone water deficits that result from the phase shift between the seasonal signals of root zone water inflow (i.e., precipitation) and outflow (i.e., evaporation). In irrigated croplands, irrigation water serves as an additional input into the root zone. However, this aspect has been ignored in many studies, and the extent to which irrigation influences $S_r$ estimates was never comprehensively quantified. In this study, our objective is to quantify the influence of irrigation on $S_r$ and identify the regional differences therein. To this aim, we integrated two irrigation methods, based on irrigation water use and irrigated area fractions, respectively, into the $S_r$ estimation. We evaluated the effects in comparison to $S_r$ estimates that do not consider irrigation for a sample of 4511 catchments globally with varying degrees of irrigation activities. Our results show that $S_r$ consistently decreased when considering irrigation with a larger effect in catchments with a larger irrigated area. For catchments with an irrigated area fraction exceeding 10 %, the median decrease of $S_r$ was 17 mm and 22 mm for the two methods, corresponding to 12 % and 17 %, respectively. $S_r$ decreased the most for catchments in tropical climates. However, the relative decrease was the largest in catchments in temperate climates. Our results demonstrate, for the first time, that irrigation has a considerable influence on $S_r$ estimates over irrigated croplands. This effect is as strong as the effects of snow melt that were previously documented in catchments that have a considerable amount of precipitation falling as snow.

## 1 Introduction

Vegetation strongly influences the water cycle as it controls the partitioning of precipitation into evaporation and discharge by transporting water from the subsurface back to the atmosphere via the roots (Milly, 1994). This water flux is defined



as transpiration and is, on average, the largest terrestrial water flux globally (Schlesinger and Jasechko, 2014). The amount
and timing of vegetation transpiration is controlled by the interplay between seasonal energy and water availability signals
(Gentine et al., 2012). The subsurface water removal by transpiration is regulated by the liquid water input and by the available
subsurface water buffer. This water buffer, the root zone storage capacity ($S_r$), is defined as the maximum volume per unit
square of subsurface moisture that is accessible to roots of vegetation for uptake (Gao et al., 2014). $S_r$ is an essential property
of hydrological systems – and parameter in land surface models and hydrological models – regulating terrestrial water, carbon
and energy balances at all scales from plot to global (Seneviratne et al., 2010; Wang and Dickinson, 2012; Dralle et al., 2020a;
Singh et al., 2022). Increasing evidence suggests that the extent of vegetation root systems, and consequently the magnitude
of $S_r$, is primarily controlled by climate conditions (Kleidon and Heimann, 1998; Gao et al., 2014; De Boer-Euser et al.,
2016). More specifically, the results of many studies suggest that the extent of root systems is a manifestation of vegetation
(i.e., the collective of all individual plants within a specified spatial domain) having efficiently adapted to past hydro-climatic
conditions. In other words, individual plants within an ecosystem have survived in competition with other plants as they found a
more efficient (or optimal) balance between above-ground and below-ground resource allocation (Kleidon and Heimann, 1998;
Collins and Bras, 2007; Guswa, 2008; Sivandran and Bras, 2013; Fan et al., 2017; Singh et al., 2020). Direct observations of
$S_r$ at scales larger than plot scale do not exist and, therefore, several indirect methods have been developed to estimate $S_r$ from
other observable ecosystem properties considering optimality principles (Kleidon, 2004; Gao et al., 2014; Speich et al., 2018;
Dralle et al., 2020a).

One of these methods is the memory method, a term coined by Van Oorschot et al. (2021), but also referred to as water
balance method (Nijzink et al., 2016; Hrachowitz et al., 2021) or mass curve technique (Gao et al., 2014; Zhao et al., 2016).
This method allows to estimate $S_r$ based on root zone water deficits arising from the phase shift between the seasonal signals
of precipitation and evaporation. This approach is based on evidence that root systems of present-day vegetation are a legacy
that reflects the memory of past water deficits during dry spells. Vegetation has efficiently adapted the extent of its root system
to past water deficits with a specific memory (i.e. the dry spell return period) to guarantee continuous access to water to
satisfy canopy water demand, but no more than that (Savenije and Hrachowitz, 2017). Numerous studies have successfully
demonstrated the potential of the memory method to provide estimates of climate controlled $S_r$ for river catchments based on
discharge data (Gao et al., 2014; De Boer-Euser et al., 2016; Van Oorschot et al., 2021), as well as on larger scales based on
remotely sensed estimates of evaporation (Wang-Erlandsson et al., 2016; Singh et al., 2020; McCormick et al., 2021; Stocker
et al., 2023). In addition, the method proved valuable to track the temporal evolution of $S_r$ due to changing hydro-climatic
conditions (Bouaziz et al., 2022) and human interventions, such as forest management (Nijzink et al., 2016; Hrachowitz et al.,
2021).

It is important to note that the memory method is based on liquid water input to the root zone. As such, solid phase precipi-
tation and storage as transient, seasonal or perennial snow packs introduces time lags between the moment of precipitation and
the release of liquid water (i.e., melt water) into the sub-surface. These time lags can lead to considerable temporal shifts in
liquid water supply, thereby affecting the development of seasonal water deficits and the associated magnitudes of $S_r$. Various
models with different levels of complexity have previously been integrated into the memory method to account for the time



lags due to snow accumulation and melt dynamics (de Boer-Euser et al., 2018; Dralle et al., 2021; Stocker et al., 2023). Dralle
et al. (2021) have recently shown that explicitly accounting for snow accumulation and associated time lags in melt water
release in the memory method does generally lead to lower values of $S_r$ in regions where significant fraction of precipitation
occurs in the form of snow.

Irrigation similarly affects the timing of water input to the soil. Besides its effect on timing, irrigation during the growing
season leads to input of additional water next to precipitation that otherwise would not be accessible for roots and thus not
be available for vegetation uptake. Irrigation thereby also affects the magnitude of water input and actively shapes the root
development of crops. Irrigation leads to shallower roots and higher root densities in the upper soil compared to non-irrigated
vegetation, as it reduces the need for resource allocation for root growth, and instead allows increased resource allocation for
above-surface growth (Klepper, 1991; Engels et al., 1994; Bakker et al., 2009; Maan et al., 2023). The strength of this sig-
nal is variable and depends on the irrigation method applied (Lv et al., 2010; Jha et al., 2017; Wang et al., 2020). Currently,
approximately 20 % of global croplands are irrigated (FAO, 2022) and with the increasing demand for crop production, irriga-
tion requirements are expected to increase in the future (Alexandratos and Bruinsma, 2012). In spite of some exceptions (e.g.
Roodari et al. (2021)), irrigation is rarely systematically represented in hydrological and biogeophysical models (McDermid
et al., 2023), mostly due to a lack of sufficient data (e.g., Meier et al. (2018)). This also holds for the memory method, as most
studies using the memory method for $S_r$ estimation did not account for irrigation, which likely led to an overestimation of $S_r$
in irrigated areas (Gao et al., 2014; De Boer-Euser et al., 2016; Stocker et al., 2023). To our knowledge, only Wang-Erlandsson
et al. (2016) explicitly accounted for irrigation when estimating $S_r$ by adding irrigation estimates simulated by the LPJmL
dynamic global vegetation model to the precipitation input (Jägermeyr et al., 2015). However, it remains unknown to which
extent irrigation influences the magnitudes of $S_r$ estimates and in which regions globally it is most relevant to take into account.

Our objective here is to quantify the influence of irrigation on the root zone storage capacity estimated with the memory
method and to identify the regional differences therein. We do so by using a sample of 4511 catchments globally with varying
degrees of irrigation activities. Specifically, we test the hypothesis that irrigation considerably reduces root zone storage capac-
ities $S_r$ and therefore needs to be accounted for in the estimation of $S_r$. To this aim, we introduce two methods that represent
irrigation based on catchment water balances to the memory method using irrigation data from two different sources. The first
method explicitly uses estimates of irrigation water use from Zhang et al. (2022) in the $S_r$ calculation with the memory method.
The second method is a simpler parameterization based on the irrigated area fraction (Siebert et al., 2015b).

## 2 Methods

### 2.1 Data

For this study we used station based discharge ($Q$) data from the following sources: the Global Streamflow Indices and Meta-
data Archive (GSIM) (Do et al., 2018; Gudmundsson et al., 2018), the Australian edition of the Catchment Attributes and
Meteorology for Large-sample Studies (CAMELS-AUS) dataset (Fowler et al., 2021), the LArge-SaMple DAta for Hydrology
and Environmental Sciences for Central Europe (LamaH-CE) (Klingler et al., 2021) and the Italian Hydrological Portal (Lend-



vai, 2020). We used annual mean discharge ($\overline{Q}$) for the catchment specific available time period. For the period 1981-2010, we obtained catchment average daily precipitation ($P$) and daily mean temperature ($T_\mathrm{a}$) from the Global Soil Wetness Project Phase 3 (GSWP3) (Dirmeyer et al., 2006) and daily potential evaporation ($E_\mathrm{p}$) from the Global Land Evaporation Amsterdam

Model version 3.5a (GLEAMv3.5a), which is based on the Priestley-Taylor approach (Martens et al., 2017; Miralles et al., 2011). We selected 4511 catchments based on the following criteria: (1) at least 10 years of $Q$ data during the 1981–2010 period; (2) catchment area $< 10000\,\mathrm{km}^2$ to limit the heterogeneity within catchments; (3) annual mean discharge ($\overline{Q}$) smaller than annual mean precipitation ($\overline{P}$) for the specific catchment.

For each catchment we obtained irrigation estimates from two different data sources. Firstly, we used the average irrigated
area fraction $I_\mathrm{a}$ (-), which is the areal fraction of land equipped with infrastructure for irrigation. $I_\mathrm{a}$ was obtained from the "AEI_HYDE_FINAL_IR" dataset developed by Siebert et al. (2015b), which is representative for the irrigation extent in the year 2005 (Fig. 1a). This dataset was based on sub-national irrigation statistics and the History Database of the Global Environment (HYDE) version 3.1 land use data (Klein Goldewijk et al., 2011; Siebert et al., 2015b). Secondly, we used estimates of annual mean irrigation water use representative for the 2011-2018 period ($\overline{I_\mathrm{w}}$ ($\mathrm{mm\,year}^{-1}$)) from Zhang et al.
(2022), who developed an algorithm to estimate irrigation from multiple satellite-based products and the Priestley-Taylor Jet Propulsion Laboratory (PT-JPL) model (Fig. 1b).

To identify the effects of irrigation for different regions, we used the Köppen-Geiger climate classes as a climate indicator. We selected for each catchment the predominant Köppen-Geiger climate class based on a global map at a $1\,\mathrm{km}$ resolution representing the 1980-2016 period (Beck et al., 2018a). The gridded data products for $P$, $E_\mathrm{p}$, $I_\mathrm{a}$ and $\overline{I_\mathrm{w}}$ were converted to
catchment estimates using area weighted averages of the grid cells that lie for more than $50\,\%$ inside the catchment. Before area weighting, the gridded products were resampled to a spatial resolution of $0.05°$ using nearest neighbour interpolation. This way, all gridded products were treated similarly and problems with small catchments with no matching grid cells were avoided.

## 2.2  Memory method with irrigation methods

Figure 2a shows a conceptualization of the memory method based on four storage components ($\mathrm{mm}$): interception storage
$S_\mathrm{i}$, snow storage $S_\mathrm{sn}$, "surplus" storage $S_\mathrm{s}$, and storage deficit $S_\mathrm{d}$. $S_\mathrm{d}$ is initially conceptualized as an infinite deficit storage volume and its temporal evolution can be described by:

$$S_\mathrm{d}(t) = \int_{t_0}^{\tau} (P_\mathrm{e} - E_\mathrm{t} + I - P_\mathrm{s})\mathrm{d}t \tag{1}$$

where $P_\mathrm{e}$ represents effective precipitation ($\mathrm{mm\,day}^{-1}$), $E_\mathrm{t}$ is transpiration ($\mathrm{mm\,day}^{-1}$), $I$ is irrigation ($\mathrm{mm\,day}^{-1}$), and $P_\mathrm{s}$ is surplus precipitation ($\mathrm{mm\,day}^{-1}$) (Fig. 2a). In Eq. (1) $t_0$ corresponds to the first day of the first hydrological year and $\tau$ to
the daily time steps ending on the last day of the last hydrological year. Our hydrological year starts the first day of the month after the wettest month, which is defined as the month with on average the largest positive difference between monthly mean $P$ and $E_\mathrm{p}$. In Eq. (1), $P_\mathrm{e}$ ($\mathrm{mm\,day}^{-1}$) is calculated from the water balance of the interception storage $S_\mathrm{i}$ (Fig. 2a), and $E_\mathrm{t}$ ($\mathrm{mm\,day}^{-1}$) is described as a fraction of daily potential evaporation $E_\mathrm{p}$ ($\mathrm{mm\,day}^{-1}$) based on the catchment water balance.



We used a simple snow model based on the degree-day method (e.g. Bergstrom, 1975; Gao et al., 2017) to account for the
delay in liquid water input to the soil by describing liquid precipitation ($P_l$ ($\mathrm{mm\,day^{-1}}$)), precipitation falling as snow ($P_{sn}$
($\mathrm{mm\,day^{-1}}$)), and snow melt ($P_m$ ($\mathrm{mm\,day^{-1}}$)). The equations for the interception storage, snow storage, and transpiration
calculation are described in Appendix A.

Surplus precipitation $P_s$ ($\mathrm{mm\,day^{-1}}$) in Eq. (1) is described by Eq. (2), in which we used the following notation for the sum
of the fluxes between two time steps: $F_t = \int_{t-1}^{t} F \mathrm{d}t$, where $F$ is either $P_e$, $E_t$, $I$ or $P_s$. Thus $P_{s,t}$ is described by:

$$P_{s,t} = \max(0, S_d + P_{e,t} - E_{t,t} + I_t),\qquad(2)$$

with $S_d$ and $I_t$ approaching zero during periods of abundant precipitation, and thus it then holds that $P_{s,t} \approx P_{e,t} - E_{t,t}$.

For the computation of applied irrigation $I$ we split the timeseries into surplus and deficit periods (Fig. 2b). For each
hydrological year, we defined one deficit period, which is the longest deficit period with the largest $S_d$ in the hydrological year.
Surplus periods were defined as the periods in between the deficit periods. For each surplus period, the surplus precipitation $P_s$
(Eq. (2)) accumulates in the surplus store $S_s$:

$$S_s(t) = \max(0, \int_{t_{s0}}^{t_{s1}} (P_s - I)\mathrm{d}t)\qquad(3)$$

with $t_{s0}$ the first day of the surplus period and $t_{s1}$ the last day of the surplus period (Fig. 2b). $S_s$ does not have a maximum
storage capacity, but it is reset to zero each year, after each deficit period. This storage conceptualizes any water buffers that can
be used for irrigation in the consecutive deficit period and may encompass ditches, lakes and aquifers. This method assumes
that irrigated water only originates from water inside the catchment boundaries and that it is sustainably extracted so that the
long term water balance is closed. During a deficit period, the fraction of $S_s$ that is used for irrigation is defined by irrigation
factor $f$ (-), which determines how much of the surplus water stored during the surplus period is used for irrigation during the
consecutive deficit period. $f$ represents both the water evaporated or discharged during the irrigation process before recharging
the soil, and the spatial extent of the irrigation. It is assumed that daily irrigation $I$ is equally distributed over the deficit period
(Fig. 2b), so that $I$ ($\mathrm{mm\,day^{-1}}$) is defined as:

$$I(t) = \frac{f S_s(t_{s1})}{\Delta t_d}\qquad(4)$$

with $\Delta t_d$ the length of the deficit period ($t_{d1} - t_{d0}$) in days (Fig. 2b). Based on the two irrigation data sources used (Sect.
2.1), we have here developed two methods to estimate $f$ in Eq. (4):

1. Irrigation Water Use method (IWU)

$f_{d,\mathrm{IWU}}$ (-) is defined for each deficit period $d$ for each catchment by:

$$f_{d,\mathrm{IWU}} = \max(1, \frac{\overline{I_w}\mathrm{d}t}{S_s(t_{s,1})}), \text{ so that } I(t) = \frac{\overline{I_w}\mathrm{d}t}{\Delta t_d} \text{ if sufficient water is available in } S_s,\qquad(5)$$

with $\overline{I_w}$ ($\mathrm{mm\,year^{-1}}$) the catchment annual mean irrigation water use, $\mathrm{d}t = 1\,\mathrm{year}$, and $S_s(t_{s,1})$ (mm) the surplus storage
at the end of the surplus storage accumulation period, i.e. the amount of water stored in $S_s$ at the start of the deficit period.





In this method, $f_{d,\mathrm{IWU}}$ is different for each deficit period $d$, as $S_\mathrm{s}$ also varies. For each catchment, $f_{\mathrm{IWU}}$ is defined as the average $f_{d,\mathrm{IWU}}$.

2. Irrigated Area Fraction method (IAF)

$f_{\mathrm{IAF}}$ (-) is temporally non-varying, and is defined for each catchment by:

$$f_{\mathrm{IAF}} = \beta I_\mathrm{a} \tag{6}$$

with $I_\mathrm{a}$ (-) the catchment irrigated area fraction and $\beta$ (-) a correction factor that is constant in space and time for all catchments. We estimated $\beta$ by minimizing the difference between $f_{\mathrm{IAF}}$ and $f_{\mathrm{IWU}}$ in terms of Root Mean Squared Error (RMSE). We generated 1000 linearly spaced values for $\beta$ between 0 and 2.5, and computed $f_{\mathrm{IAF}}$ for all the catchments. For all these cases, the RMSE of catchment $f_{\mathrm{IAF}}$ and $f_{\mathrm{IWU}}$ was computed (Fig. 3). The RMSE minimized for $\beta = 0.9$ (RMSE $= 0.044$), which is applied for all catchments in Eq. (6).

To evaluate the effect of these methods on estimated $S_\mathrm{r}$ we tested a third case, referred to as No Irrigation (NI), in which $f_{\mathrm{NI}} = 0$. A priori we cannot and do not consider any of the two methods, i.e., the IWU or the IAF method to be more representative than the other. While the IWU method uses the irrigation data more directly than the IAF method, the latter directly takes the inter-annual variability of surplus water into account.

### 2.2.1 Root zone storage capacity calculation

Catchment-scale root zone storage capacity $S_\mathrm{r}$ was here derived from the catchment-scale storage deficit $S_\mathrm{d}$ timeseries for the three different irrigation cases NI, IWU, and IAF (Table 1). For each catchment, the annual maximum storage deficits ($S_{\mathrm{d,M}}$) were defined for each hydrological year as:

$$S_{\mathrm{d,M}} = \max(S_\mathrm{d}(t)) - \min(S_\mathrm{d}(t)) \tag{7}$$

with the $\min(S_\mathrm{d})$ occurring earlier in the hydrological year than the $\max(S_\mathrm{d})$. Previous studies (e.g. Gao et al., 2014; Wang-Erlandsson et al., 2016) applied a Gumbel distribution on the $S_{\mathrm{d,M}}$ values to estimate $S_\mathrm{r}$ for different return periods $T$. For croplands, and thus irrigated land, a return period of 2 years was found to the be the most representative (Wang-Erlandsson et al., 2016). Here, we directly used the observed $S_{\mathrm{d,M}}$ values with occurrences closest to $T = 2$ years instead of a fitted extreme value distribution, because fitting an extreme value distribution is ambiguous for return periods of interest (here: 2 years) much smaller than the timeseries length (here: >10 years). For all catchments, the $S_\mathrm{r}$ was estimated as the mean of the three observed $S_{\mathrm{d,M}}$-values with occurrences closest to $T = 2$ years.

### 2.2.2 Evaluation

To visualize the effects of irrigation on $S_\mathrm{d}$ and $S_\mathrm{r}$, we selected four example catchments with different irrigation magnitudes (i.e., $I_\mathrm{a}$ and $\overline{I_\mathrm{w}}$) in four different continents and climate zones. For quantification of the effects of irrigation on $S_\mathrm{r}$, we computed





absolute ($\Delta$) and relative ($\Delta_\mathrm{r}$) differences between the $S_\mathrm{r}$ estimates for the NI, IWU and IAF cases (Table 1). Catchments were stratified based on (1) four different ranges of irrigated area $I_\mathrm{a}$: $I_\mathrm{a} \leq 0.01$; $0.01 < I_\mathrm{a} \leq 0.05$; $0.05 < I_\mathrm{a} \leq 0.1$; and $I_\mathrm{a} > 0.1$

(Fig. S2); (2) regions, i.e., South-America, North-America, Europe and Asia; and (3) climate zones based on Köppen-Geiger, subdivided into Tropical (Af, Am, Aw), Arid (BWh, BWk, BSh, BSk), Temperate (Cfa, Cfb, Cfc), Mediterranean (Csa, Csb), and Continental (Dfa, Dfb, Dfc, Dfd) climates, with the abbrevations of the Köppen-Geiger climate classification (Beck et al., 2018a) (Fig. S3). Uncertainty of the differences in $S_\mathrm{r}$ were represented by the interquartile range (IQR).

## 3 Results

### 190 3.1 Irrigation influence on root zone storage capacity

Globally, the $S_\mathrm{r}$ estimates without accounting for irrigation ranged from 0–800 mm, with larger values in semi-arid regions with high rainfall seasonality such as North-Eastern Brazil (median $S_\mathrm{r} \approx 250$ mm) or monsoon regions such as North-Eastern Indian (median $S_\mathrm{r} \approx 450$ mm), than in regions with temperate climates with year-round rainfall such as Western Europe (median $S_\mathrm{r} \approx 70$ mm) or continental, colder, climates such as Canada (median $S_\mathrm{r} \approx 40$ mm) (Fig. 4).

The storage deficits $S_\mathrm{d}$ (Eq. 1) in general reduced when accounting for irrigation effects according to the IWU and IAF cases as compared to the case without irrigation (NI). These overall effects of the method are illustrated by four selected example catchments in Fig. 5. More pronounced effects of irrigation on $S_\mathrm{d}$ are visible for the example catchments in Europe (Fig. 5e,f) and Asia (Fig. 5g,h), with larger $\overline{I_\mathrm{w}}$ and $I_\mathrm{a}$, than in the example catchments in South-America (Fig. 5a,b) and North-America (Fig. 5c,d). As $S_\mathrm{d}$ decreased, the annual maximum storage deficits $S_\mathrm{d,M}$, as determined by Eq. (7), decreased as

well. Consequently, the estimated $S_\mathrm{r}$ decreased for the IWU and IAF cases compared to NI, with more pronounced effects in the example catchments with larger $\overline{I_\mathrm{w}}$ and $I_\mathrm{a}$ (Fig. 5). Globally, $S_\mathrm{r}$ consistently decreased for IWU and IAF (Fig. 6), albeit the magnitudes vary to a considerable extent. Nevertheless, relatively clear regional patterns of the effects of irrigation on $S_\mathrm{r}$ emerged. The most pronounced effects cluster in catchments in regions that are characterized by widespread and intense crop cultivation, and thus high irrigation water use, such as Northern Spain and France, Central US and parts of India (Fig. 1).

### 205 3.2 Regional differences of irrigation influence on root zone storage capacity

Figure 7 shows that the effects of irrigation on $S_\mathrm{r}$ increased with increasing irrigated area fraction $I_\mathrm{a}$ for both IWU and IAF cases. We found the largest effects in catchments with $I_\mathrm{a} > 0.1$, such as the example catchment in Asia (Fig. 5g). For these catchments, the median $\Delta S_\mathrm{r}$ was 18 mm (IQR 9–29 mm) for IWU and 20 mm (IQR 9–42 mm) for IAF (Fig. 7), which correspond with decreases of 12 % and 17 %, respectively (Table 2). These effects were considerably larger than the effects of

210 irrigation in catchments with $0.05 < I_\mathrm{a} \leq 0.1$ that reached median $\Delta_\mathrm{r} S_\mathrm{r}$ of 6–7%, which corresponds to $\Delta S_\mathrm{r} \approx 9$ mm (Fig. 7, Table 2). Although the median effects of irrigation on $S_\mathrm{r}$ for catchments with $I_\mathrm{a} \leq 0.05$ were relatively small, the effects can be considerable for specific individual catchments as shown by the outliers in Fig. 7.



The strongest irrigation influence on $S_r$ for catchments with $I_a > 0.05$ was found in Asia, followed by South-America, Europe, and North-America for both IWU and IAF (Fig. 8a). For the catchments in Asia we found median values of $\Delta S_r$ for

IWU of 21 mm (IQR 13–41 mm), and for IAF of 27 mm (IQR 12–56 mm). However, the relative differences in $S_r$ were with $\Delta_r S_r$ =9–10 % smaller in Asia than in other regions, reaching up to 24 % in North America, because the initial $S_r$ without accounting for irrigation was considerably larger in Asia than in other regions (Fig. 4, Table 2). Figure 8b shows that $S_r$ decreased the most in tropical catchments with median $\Delta S_r = 19$ mm for IWU and 24 mm for IAF. These findings are in line with the results presented in Fig. 8a since most of the tropical catchments we evaluated were located in Asia (Fig. S3).

For catchments in the arid, Mediterranean, temperate, and continental climate zones, median $\Delta S_r$ was comparable and varied between 4 mm and 11 mm. However, catchments in temperate climates exhibited the largest relative influence of irrigation on $S_r$ with median $\Delta_r S_r$ =16 % for IWU and 22 % for IAF (Table 2).

### 3.3 Comparison IWU and IAF methods

Figure 6 shows similar spatial patterns of $\Delta_r S_r$ for IWU and IAF, but the magnitudes differed. For most groups of catchments,

IAF had a more pronounced effect on $S_r$ than IWU (Table 2). The different results for IWU and IAF can be explained by the different methodologies (Table 1). The IWU method directly used annual mean irrigation water use $\overline{I_w}$ from Zhang et al. (2022) as an estimate for $I$, if sufficient water was available in the surplus store $S_s$. On the other hand, in the IAF method $I$ was defined as a fraction of $S_s$ based on the irrigated area fraction $I_a$ and the constant $\beta$. Therefore, the estimated $I$ in IAF directly reflected the inter-annual variability of surplus water. Another cause for the different results for IWU and IAF lies in

the estimation of $\beta$ in IAF, which was based on minimization of the differences between $f_{i,IWU}$ and $f_{i,IAF}$ (Sect. 2.2; Fig. 3). In spite of this optimization, differences between $f_{i,IWU}$ and $f_{i,IAF}$ remained, which partially explain the differences in $\Delta S_r$ between the two methods.

## 4 Discussion

### 4.1 Synthesis of results

Our results showed that the effect of irrigation on $S_r$ is discernible in all regions, but the magnitude of the effect depends on the amount of irrigation applied (Fig. 5–8). For many parts of the world the integration of irrigation in the $S_r$ estimation did not have a large influence (Fig. 6). However, $S_r$ considerably reduced for catchments with irrigated area fractions $I_a > 0.05$, and ignoring irrigation in these regions would lead to biased estimates of $S_r$, and, as a consequence, to inadequate modeling of vegetation transpiration (Fig. 7). The influence of irrigation on $S_r$ estimates, as presented in Fig. 6, resembled the spatial

pattern found in global assessments of irrigation water withdrawal (Huang et al., 2018), and the extent of irrigation activities (McDermid et al., 2023). This was expected since we used similar underlying irrigation data in the here developed irrigation methods. Previous studies using the memory method did not consider irrigation in $S_r$ estimates (e.g. De Boer-Euser et al., 2016; Stocker et al., 2023), or did not evaluate its effects (Wang-Erlandsson et al., 2016). To put our results into perspective,





we looked at the effects of snow accumulation and melt on $S_r$ estimates for the continental United States from Dralle et al.
(2021), as this process alters the $S_d$ time series in a similar way as irrigation by temporally shifting liquid water input into the
system. Dralle et al. (2021) estimated that integrating snow accumulation and melt in the memory method led to an average
reduction in $S_r$ of 6 mm (2 %) for areas with >10 % winter snow coverage, and 28 mm (17 %) for areas with >80 % winter
snow coverage (Dralle et al., 2020b). These magnitudes are broadly consistent with our findings for irrigation (Table 2).

Both the results of IWU and IAF showed considerable effects of irrigation on $S_r$ (Figs. 6–8), and both are suitable to use in
the memory method, keeping in mind the individual uncertainties related to data and methodological assumptions. We think
that the IWU method is more suitable for regional application for periods with available $I_w$ data (Zhang et al., 2022) than IAF,
because $I_w$ was derived from water balances, that strongly depend on the evaluated period. However, for spatial and temporal
extrapolation the direct use of the $I_w$ data in the IWU method is more uncertain than the simpler IAF method, because the
irrigated area fraction $I_a$ used in IAF is expected to be temporally less variable than the water used for irrigation $I_w$. Therefore,
we think the simpler parameterized IAF method is more suitable to use in the memory method for global applications and
varying time periods. Moreover, IAF has the potential to be integrated dynamically in hydrological or land surface models
used for global Earth system model studies and future predictions.

## 4.2 Methodological limitations

By using several data sources, we obtained a large sample of 4511 catchments on different continents, characterized by a wide
spectrum of climates, and in particular, regions with various levels of irrigation activity. However, the global coverage is not
entirely balanced as Africa and large parts of Asia were undersampled. A further limitation may arise from the assumption in
the memory method with irrigation methods proposed here that catchments are hydrologically closed systems. However, inter-
catchment lateral flows, such as groundwater and irrigation water can significantly alter catchment water balances (e.g., Bouaziz
et al., 2018; Fan, 2019; Condon et al., 2020). Moreover, the extraction of fossil groundwater for irrigation (Siebert et al., 2010;
Grogan et al., 2017; de Graaf et al., 2019) can violate the assumption of closing water balances for the here developed irrigation
methods in the memory method. Another limitation was the availability and quality of irrigation data (Sect. 2.1, Fig. 1). The
annual mean $\overline{I_w}$ used in IWU was based on the 2011-2018 period, while the catchment time series varied between 1981 and
2010. Similarly, the $I_a$ we used represented the 2005 irrigated area fraction (Siebert et al., 2015b). The temporal mismatch
may have led to an overestimation of $I$ for the specific period, as irrigated area, and irrigation techniques and efficiency have
developed over the evaluated period (McDermid et al., 2023). However, this did not have major influence the quantification of
the general patterns of the effects of irrigation on $S_r$, which was the aim of this study.

An additional source of uncertainty in the application of the memory method, as used in this study, relates to the derivation
of $S_r$ from the $S_d$ time series (Fig. 5). Given that an ecosystem has developed its $S_r$ in a way it optimally functions and can
overcome dry periods (e.g. Guswa, 2008), $S_r$ for a specific time period would correspond to the maximum $S_d$ value observed
during that same time period ($S_r = \max(S_d)$). However, it is important to note that the memory method represents a simplified
approximation of real ecosystem behavior and has inherent limitations. The most important limitation is that our application of
the memory method did not account for the feedback between $S_d$ and $E_t/E_p$, which likely led to an overestimation of $S_d$ (Van





Oorschot et al., 2021). In this study, we primarily focused on crops, that do not exhibit a multi-year root adaptation for survival, as there is no remaining $S_r$ after each year's harvest. However, the catchments used here are in no case entirely covered by
280 crops, and, therefore, we used a return period of 2 years for the $S_r$ estimation (following Wang-Erlandsson et al. (2016)).

## 5   Conclusions

Using a large sample of catchments globally, the presented results support the hypothesis that irrigation considerably reduces root zone storage capacity $S_r$ estimated with the memory method. We found a median reduction of $S_r$ by 12 % (IQR 7–23 %) for the IWU method and 17 % (IQR 6–36 %) for the IAF method for catchments with an irrigated area fraction $I_a >10$ %.
In general, these effects were less pronounced in catchments with a smaller irrigated area, although the $S_r$ for individual catchments could also be considerably influenced by irrigation. $S_r$ decreased the most for catchments in tropical climates with a median decrease of 19–24 mm (for $I_a >5$ %). The reductions of $S_r$ found in this study are in the same order of magnitude as the snow effects on $S_r$ estimated by Dralle et al. (2021). Of paramount relevance for regional-to-global hydrological and climate modelling studies, this study demonstrates the relevance of irrigation for adequately estimating $S_r$. The irrigation water
use can be expected to further increase over the next decades and so the related effects on $S_r$ should be represented in the Earth system models that are used for the next climate projections. The methodological approach developed in this study could be profitably used in this respect.

*Code and data availability.*  GSIM discharge data was obtained from https://doi.pangaea.de/10.1594/PANGAEA.887477 (Do et al., 2018) and https://doi.pangaea.de/10.1594/PANGAEA.887470 (Gudmundsson et al., 2018), CAMELS-AUS data from https://doi.pangaea.de/10.
1594/PANGAEA.921850 (Fowler et al., 2020), LamaH-CE data from https://zenodo.org/record/7691294 (Kauzlaric et al., 2023), discharge data for Italian catchments from http://meteoniardo.altervista.org/ (Lendvai, 2020). GSWP3 precipitation and daily mean temperature were obtained from https://data.isimip.org/search/simulation_round/ISIMIP2a/product/InputData/climate_forcing/gswp3/ (Lange and Büchner, 2020) and potential evaporation from GLEAM v3.5a was downloaded from https://www.gleam.eu/#downloads (Miralles et al., 2011; Martens et al., 2017). Irrigated area fraction was downloaded from https://mygeohub.org/publications/8/2 (Siebert et al., 2015a) and irrigation water
use from https://data.tpdc.ac.cn/en/data/ad558954-bc55-44cf-94c8-bdb33820f784/ (Xin et al., 2021). The global map of the Köppen-Geiger climate classification was obtained from https://figshare.com/articles/dataset/Present_and_future_K_ppen-Geiger_climate_classification_maps_at_1-km_resolution/6396959/2 (Beck et al., 2018b). Upon acceptance, the data and scripts underlying this manuscript will be shared in an open access research database.

## Appendix A:  Memory method equations

These equations follow Van Oorschot et al. (2021), who based their methods on Gao et al. (2014); De Boer-Euser et al. (2016); Nijzink et al. (2016) and Wang-Erlandsson et al. (2016). Total precipitation ($P$ (mm day$^{-1}$)) was split into liquid precipitation ($P_l$ (mm day$^{-1}$)), and precipitation falling as snow ($P_{sn}$ (mm day$^{-1}$)) based on temperature (Fig. 2). As temperature varies



with altitude, we divided each catchment in elevation zones of $250\,\mathrm{m}$. For each elevation zone $z$ the daily temperature $T_z$ was calculated by:

$$T_z(t) = T_\mathrm{a}(t) + \lambda \Delta H \tag{A1}$$

with $T_\mathrm{a}$ ($^\circ$C) the catchment average temperature (GSWP3), $\lambda$ the lapse rate of $0.0064\,^\circ\mathrm{C}\,\mathrm{m}^{-1}$, and $\Delta H$ (m) the elevation difference between the elevation zone and the mean elevation. For each elevation zone $z$ daily $P_\mathrm{l}$ and $P_\mathrm{sn}$ were defined by:

$$P_{\mathrm{l},z}(t) = \begin{cases} P(t) & \text{if } T_z(t) > T_\mathrm{t} \\ 0 & \text{if } T_z(t) < T_\mathrm{t} \end{cases} \tag{A2}$$

$$P_{\mathrm{sn},z}(t) = \begin{cases} P(t) & \text{if } T_z(t) < T_\mathrm{t} \\ 0 & \text{if } T_z(t) > T_\mathrm{t} \end{cases} \tag{A3}$$

The water balance of the snow storage ($S_\mathrm{sn}$) (Fig. 2) for each elevation zone $z$ was described by:

$$\frac{\mathrm{d}S_{\mathrm{sn},z}}{\mathrm{d}t} = P_{\mathrm{sn},z} - P_{\mathrm{m},z}. \tag{A4}$$

Equation (A4) can be solved by Eqs. (A5) and (A6), in which we used for the sum of fluxes between two time steps the following notation: $F_t = \int_{t-1}^{t} F \mathrm{d}t$, where $F$ is either $P_\mathrm{sn}$ or $P_\mathrm{m}$. Thus, the numerical solution using daily time steps can be described as:

$$S_{\mathrm{sn},z,t} = S_{\mathrm{sn},z,t-1} + P_{\mathrm{sn},z,t} - P_{\mathrm{m},z,t} \tag{A5}$$

$$P_{\mathrm{m},z,t} = \begin{cases} \max(M(T_{z,t} - T_\mathrm{t}),\ S_{\mathrm{sn},z,t}) & \text{if } T_{z,t} < T_\mathrm{t} \\ 0 & \text{if } T_{z,t} > T_\mathrm{t} \end{cases} \tag{A6}$$

with $T_\mathrm{t}$ the threshold temperature for snowfall of $0\,^\circ$C, and $M$ the snow melt factor of $2\,\mathrm{mm}\,\mathrm{d}^{-1}\,^\circ\mathrm{C}^{-1}$. Total catchment $P_\mathrm{l}$, $P_\mathrm{sn}$, and $P_\mathrm{m}$ were calculated as an area weighted sum of the values for the different elevation zones.

The calculation of effective precipitation $P_\mathrm{e}$ ($\mathrm{mm}\,\mathrm{day}^{-1}$) and transpiration $E_\mathrm{t}$ in Eq. (1) are similar as in Van Oorschot et al. (2021). The water balance of the interception store $S_\mathrm{i}$ (Fig. 2a) is described by:

$$\frac{\mathrm{d}S_\mathrm{i}}{\mathrm{d}t} = P_\mathrm{l} - P_\mathrm{e} - E_\mathrm{i} \tag{A7}$$

with $P_\mathrm{l}$ the liquid precipitation ($\mathrm{mm}\,\mathrm{day}^{-1}$) and $E_\mathrm{i}$ the interception evaporation ($\mathrm{mm}\,\mathrm{day}^{-1}$). Equation (A7) can be solved by Eqs. (A8)-(A10), in which we used for the sum of fluxes between two time steps the following notation: $F_t = \int_{t-1}^{t} F \mathrm{d}t$, where



$F$ is either $P_\mathrm{l}$, $E_\mathrm{i}$, $P_\mathrm{e}$ or $E_\mathrm{p}$ (potential evaporation ($\mathrm{mm\,day}^{-1}$)). Thus, the numerical solution using daily time steps can be described as:

$$P_{\mathrm{e},t} = \begin{cases} 0 & \text{if } P_{\mathrm{l},t} + S_{\mathrm{i},t-1} \leq S_{\mathrm{i,max}} \\ P_{\mathrm{l},t} + S_{\mathrm{i},t-1} - S_{\mathrm{i,max}} & \text{if } P_{\mathrm{l},t} + S_{\mathrm{i},t-1} > S_{\mathrm{i,max}} \end{cases} \tag{A8}$$

$$S^*_{\mathrm{i},t} = S_{\mathrm{i},t-1} + P_{\mathrm{l},t} - P_{\mathrm{e},t} \tag{A9}$$

$$E_{\mathrm{i},t} = \begin{cases} E_{\mathrm{p},t} & \text{if } E_{\mathrm{p},t} < S^*_{\mathrm{i},t} \\ S^*_{\mathrm{i},t} & \text{if } E_{\mathrm{p},t} \geq S^*_{\mathrm{i},t} \end{cases} \tag{A10}$$

with $E_\mathrm{p}$ potential evaporation ($\mathrm{mm\,day}^{-1}$) and $S_{\mathrm{i,max}}$ the maximum interception storage (mm). The size of $S_{\mathrm{i,max}}$ has minor influence on estimates of $S_\mathrm{r}$ as shown by e.g. Hrachowitz et al. (2021) and Bouaziz et al. (2020), and was, therefore, set to a

constant value of $2.5\,\mathrm{mm}$. Daily transpiration ($E_\mathrm{t}$) in Eq. (1) was calculated as a fraction of daily $E_\mathrm{p}$ by:

$$E_\mathrm{t} = (E_\mathrm{p} - E_\mathrm{i}) \frac{\overline{E_\mathrm{t}}}{\overline{E_\mathrm{p}} - \overline{E_\mathrm{i}}} \tag{A11}$$

with $\overline{E_\mathrm{t}}$ the long-term mean $E_\mathrm{t}$ derived from the water balance ($\overline{E_\mathrm{t}} = \overline{P_\mathrm{e}} - \overline{Q}$), and $\overline{E_\mathrm{p}}$ the long-term mean $E_\mathrm{p}$.

*Author contributions.* FO conceptualized the study and carried out the formal analysis with input from RE and MH. FO prepared the manuscript with contributions from RE, MH and AA.

*Competing interests.* MH is member of the Editorial Board of HESS. The authors have no other competing interests to declare.

*Acknowledgements.* This work was supported by Netherlands Organization for Scientific Research (NWO) under grant OCENW.XS22.2.109, and by the European Union's Horizon 2020 research and innovation program under grant agreement no. 101004156 (CONFESS project). Acknowledgement is given for the use of the DelftBlue computing facility at the Delft High Performance Computing Center (DHPC) (Delft

High Performance Computing Centre , DHPC).





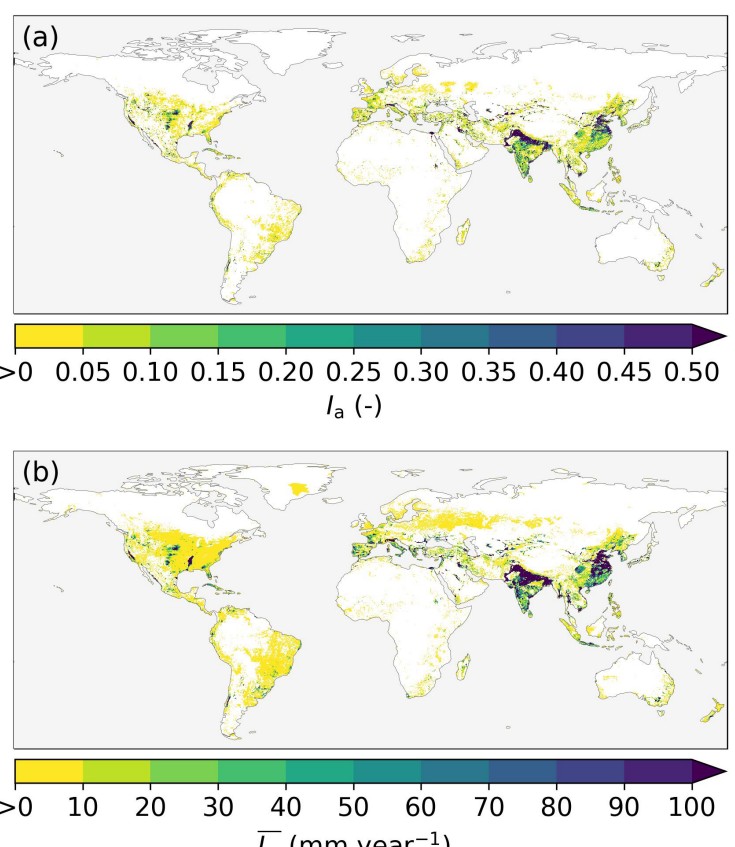

**Figure 1.** Global irrigation characteristics. (a) Irrigated area fraction ($I_a$ (-)) representative for 2005 based on sub-national irrigation statistics and the HYDE 3.1 land use data (Siebert et al., 2015b). (b) Annual mean irrigation water use ($\overline{I_w}$ (mm year$^{-1}$)) for the period 2011-2018 based on multiple satellite-based products and the PT-JPL model (Zhang et al., 2022). White areas indicate $I_a$ or $\overline{I_w}$ equal to zero.





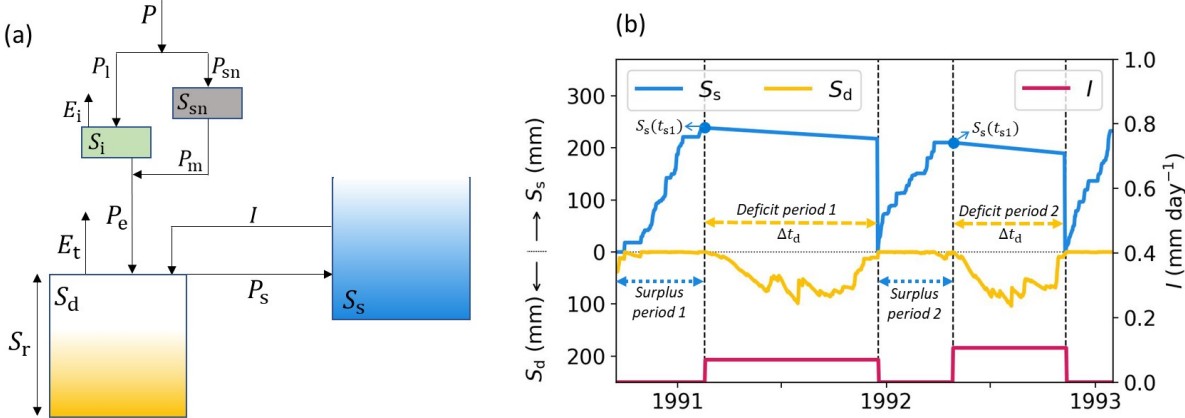

**Figure 2.** (a) Schematic bucket model representation of the memory method including an irrigation model with the following storages (mm): interception storage ($S_\mathrm{i}$), snow storage ($S_\mathrm{sn}$), storage deficit ($S_\mathrm{d}$), surplus storage ($S_\mathrm{s}$), and root zone storage capacity ($S_\mathrm{r}$); and fluxes (mm day$^{-1}$): total precipitation ($P$), liquid precipitation ($P_\mathrm{l}$), precipitation falling as snow ($P_\mathrm{sn}$), interception evaporation ($E_\mathrm{i}$), snow melt ($P_\mathrm{m}$), effective precipitation ($P_\mathrm{e}$), transpiration ($E_\mathrm{t}$), precipitation surplus ($P_\mathrm{s}$) and irrigation ($I$). (b) An example time series of $S_\mathrm{s}$, $S_\mathrm{d}$ and $I$ based on Eqs. (1–6), with $\Delta t_\mathrm{d}$ the length of the deficit period (days), and $S_\mathrm{s}(t_\mathrm{s1})$ the surplus storage at the end of the surplus period.





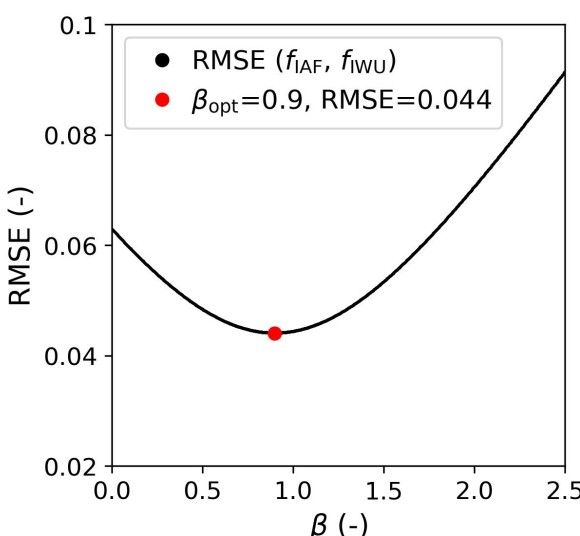

**Figure 3.** Root Mean Squared Error (RMSE) between the catchment irrigation factors $f_{\mathrm{IWU}}$ (Eq. 5) and $f_{\mathrm{IAF}}$ (Eq. 6) for 4511 catchments for 1000 linearly spaced values of $\beta$ between 0 and 2.5. $\beta_{\mathrm{opt}}$ represents the value for $\beta$ where the RMSE minimizes.



**Table 1.** Details of the irrigation cases considered in this study.

| Irrigation case | Details | Irrigation factor $f$ (Eq. (4)) |
|---|---|---|
| NI | No irrigation | $f_{\mathrm{NI}} = 0$ |
| IWU | Irrigation based on Irrigation Water Use (Fig. 1b) | $f_{\mathrm{IWU}} = \max(1, \frac{\overline{I_{\mathrm{w}}} \mathrm{d}t}{S_{\mathrm{s}}(t_{s,1})})$ (Eq. (5)) |
| IAF | Irrigation based on Irrigated Area Fraction $I_{\mathrm{a}}$ (Fig. 1a) | $f_{\mathrm{IAF}} = \beta I_{\mathrm{a}}$ (Eq. (6)) |





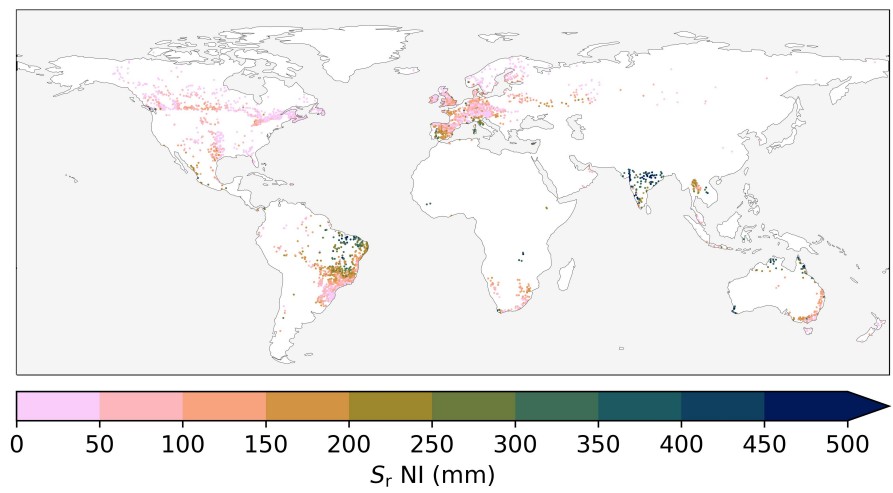

**Figure 4.** Catchment $S_r$ for the No Irrigation (NI) case, with dots representing catchment outlets. Similar figures for the IWU and IAF cases are presented in Fig. S1.





**Figure 5.** (a, c, e, g) Timeseries of storage deficits $S_d$ (mm) (Eq. 1) for four illustrative catchments with increasing irrigation from top to bottom for the three irrigation cases NI, IWU, and IAF (Table 1) with for each catchment the associated annual mean irrigation water use ($\overline{I_w}$), irrigated area fraction ($I_a$), and root zone storage capacity ($S_r$) values. (b, d, f, h) Return level plot of annual maximum storage deficits ($S_{d,M}$) (Eq. 7) for the three irrigation cases NI, IWU and IAF with the dashed vertical line corresponding to a return period $T$ of 2 years (Section 2.2.1). The locations of the catchments are shown in Fig. 6. Catchment identity, continent, and Köppen Geiger climate zone are from top to bottom: br_0002356, South America, temperate (Cfb); ca_0000689, North America, continental (Dfb); es_0000742, Europe, Mediterranean (Csa); in_0000252, Asia, tropical (Aw).



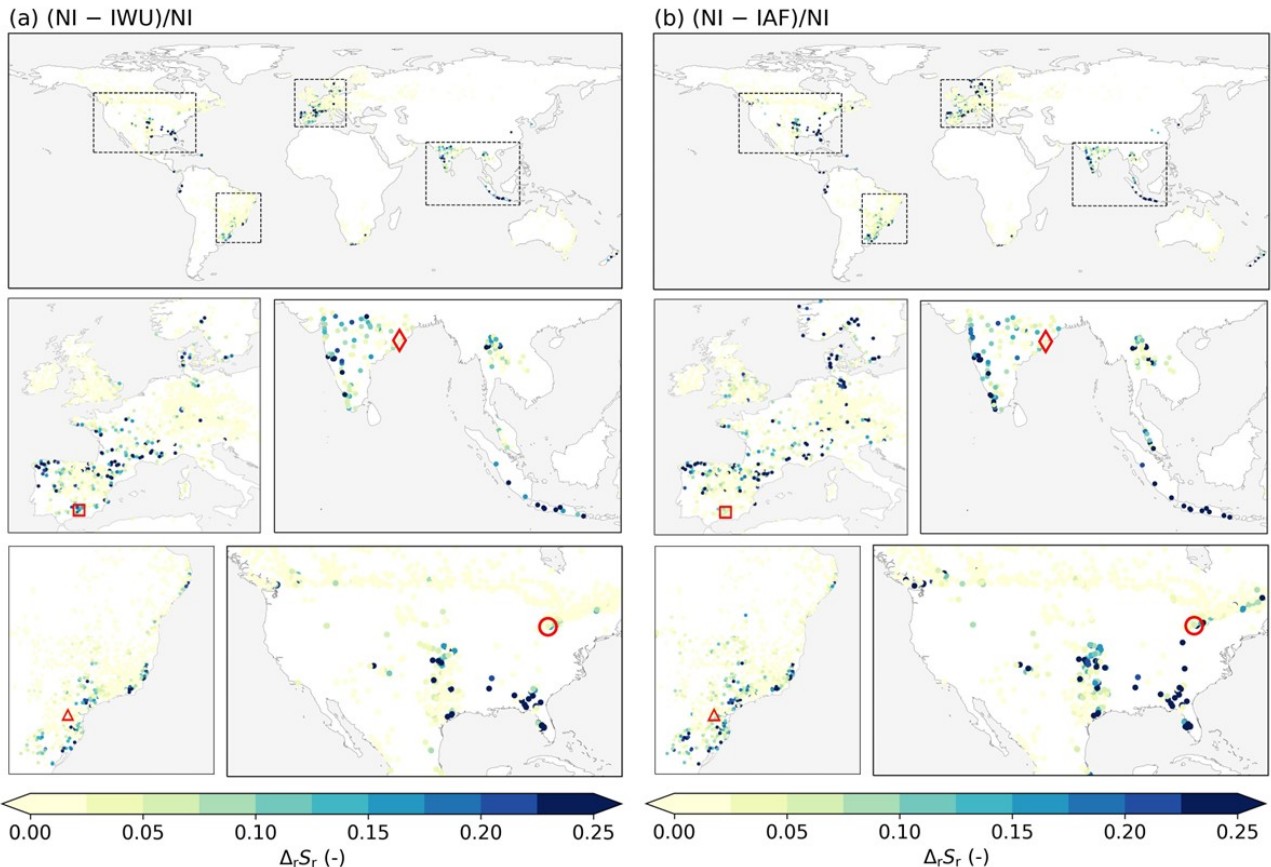

**Figure 6.** Relative difference in $S_r$ ($\Delta_r S_r$ (-)) for (a) IWU compared to NI ((NI-IWU)/NI) and (b) IAF compared to NI ((NI-IAF)/NI). Red markers indicate the selected catchments from Fig. 5. See Table 1 for details on the irrigation cases.





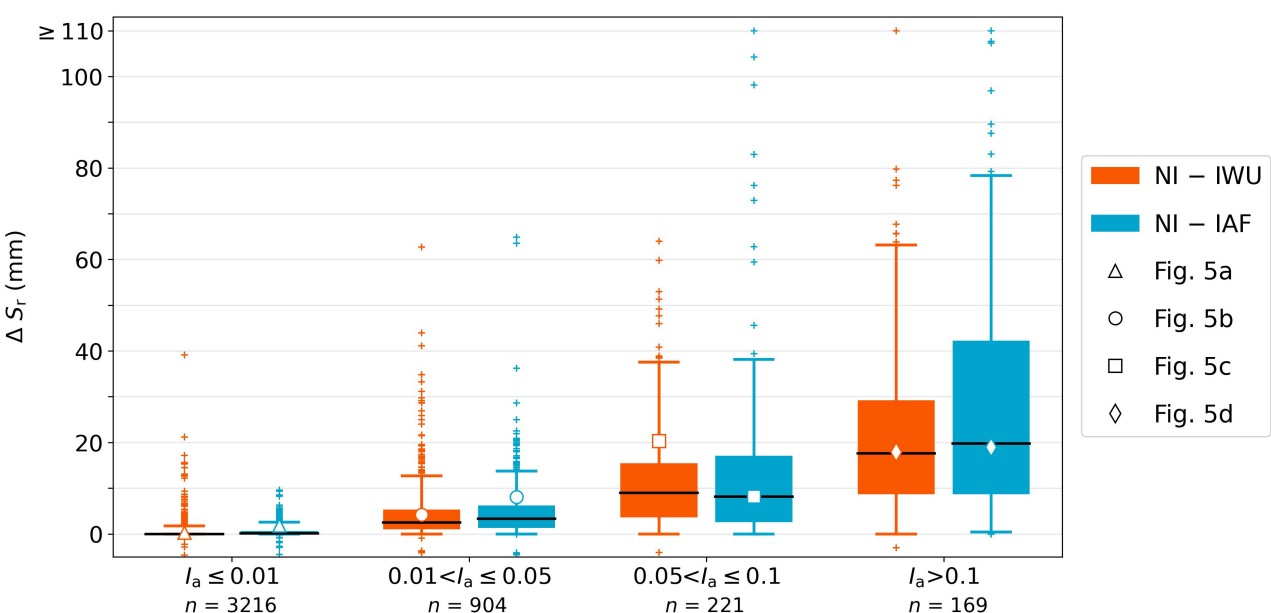

**Figure 7.** Boxplots of absolute $S_r$ difference ($\Delta Sr\,(\mathrm{mm})$) between the irrigation cases (IWU and IAF) and the no irrigation case (NI) (Table 1). Catchments are stratified in four groups based on the irrigated area fraction $I_a$ (Fig. S2), with $n$ the number of catchments in each group. The black line represents the median, the box the interquartile range (IQR), and the whiskers the 5th and 95th percentiles. White markers represent the points presented in Fig. 5. Median and IQR values for relative $S_r$ differences ($\Delta_r Sr\,(\%)$) are presented in Table 2.






**Figure 8.** Boxplots of absolute $S_r$ difference ($\Delta S_r$ (mm)) between the irrigation cases (IWU and IAF) and the no irrigation case (NI) (Table 1). In (a) catchments are stratified regionally, similar to the maps in Fig. 6, and in (b) catchments are stratified based on climate zone (Sect. 2.2.2, Fig. S3), with for both (a) and (b) only catchments with irrigated area fraction $I_a > 0.05$. The total number of catchments ($n$) in each group is given, with the numbers in brackets representing $n$ in $0.05 < I_a \leq 0.1$, and $I_a > 0.1$, respectively. The black line represents the median, the box the interquartile range (IQR), and the whiskers the 5th and 95th percentiles. Median and IQR values for relative $S_r$ differences ($\Delta_r S r$ (%)) are presented in Table 2.





**Table 2.** Median and interquartile range (IQR) of the relative $S_r$ difference ($\Delta_r Sr\,(\%)$) between the irrigation cases (IWU and IAF) and the no irrigation case (NI) with the catchments stratified for the top four rows based on irrigated area fraction ($I_a$) (Fig. 7), for the middle four rows based on region (Fig. 8a), and for the bottom five rows based on climate zones (Fig. 8b). IQR is given as the 25th percentile – 75th percentile.

|  | (NI–IWU)/NI | | (NI–IAF)/NI | |
|---|---|---|---|---|
|  | median | IQR | median | IQR |
| $I_a \leq 0.01$ | 0 | 0–0 | 0 | 0–1 |
| $0.01 < I_a \leq 0.05$ | 2 | 1–6 | 3 | 1–8 |
| $0.05 < I_a \leq 0.1$ | 6 | 3–15 | 7 | 2–21 |
| $I_a > 0.1$ | 12 | 7–23 | 17 | 6–36 |
| South-America | 12 | 7–20 | 14 | 8–29 |
| North-America | 11 | 4–28 | 24 | 6–33 |
| Europe | 9 | 4–19 | 7 | 2–26 |
| Asia | 9 | 5–16 | 10 | 4–21 |
| Tropical | 9 | 5–17 | 10 | 5–26 |
| Arid | 5 | 2–11 | 3 | 1–7 |
| Mediterranean | 5 | 0–10 | 3 | 1–7 |
| Temperate | 16 | 7–28 | 22 | 7–35 |
| Continental | 9 | 4–17 | 26 | 6–36 |



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
