# Peer review of "Influence of irrigation on root zone storage capacity estimation"

_EGUsphere, 2023_

## Author Comment (AC1)

**Reviewer 1**

*This paper assesses the impact of accounting for irrigation when calculating root zone storage based on a wealth of large hydroclimatic datasets applied to numerous catchments across the globe. The authors evidence a general reduction of root zone storage estimates, more marked in absolute value in locations with more than 10% of irrigated area (17-22mm reduction) or in a humid tropical climate, while the relative reduction is largest in temperate locations (16-22% reduction).*

*In revising the estimates plant-accessible storage estimates in irrigated crops, this paper tackles an issue of key importance for estimating current and future ecohydrological feedback in the Earth System. It is therefore an important step, and I found the manuscript pleasant to read, concise and clear for the most part. In my view it may actually a bit too concise, lacking contextualization through a more in-depth discussion. After this and some clarifications, I think it will be suitable for publication in HESS.*

We would like to thank the referee for the comments. We appreciate the time and effort taken to read our manuscript in detail and to provide us the very useful and interesting thoughts on our research. We will take the comments into account when revising the manuscript.

We have separated the different comments (shown in *italic*) and have written our replies below. Text in the original manuscript is shown in '*italic*' and revised text in '**bold**'. Unless differently stated, line numbers mentioned in our reply refer to the original manuscript version.

**General comments**

*The discussion is quite short, and a significant part of it is a synthesis of the results, I think it could dig deeper in the implications and robustness of the method and results. These could be (but not limited to):*

**Comment 1.1**

*The comparison with the impact of using snow accumulation is very interesting, all the more that the present study also considers snow storage. It would be quite interesting to see the relative effects of snow and irrigation in catchments where both are significant, with a "no-snow" case (e.g. by forcing P_sn to zero).*

We have 27 catchments where both snow (snow days > 5% of the total days) and irrigation ($I_a$>0.05) are considerable. For these catchments we found an average reduction of $S_r$ between the no irrigation-no snow case (NI_NS) and the no irrigation-snow case (NI_S) of 7 mm (-7 %) (Table 1). Including irrigation with IWU and IAF leads to a further $S_r$ reduction of 6 mm (7%) for IWU, and 11 mm (12%) for IAF. Based on these numbers we can conclude that the relative effects of irrigation and snow are also comparable within our study.

*Table 1. Average $S_r$ results for the 27 catchments with snow and irrigation significant ($I_a$>0.05) with the following cases: NI_NS: no irrigation and no snow; NI_S: no irrigation and snow; IWU_S: irrigation based on irrigation water use and snow; IAF_S: irrigation based on irrigated area fraction and snow. The right side of the table shows the absolute and relative differences between the mean Sr values.*

| | Mean $S_r$ | | | Absolute difference | Relative difference |
|---|---|---|---|---|---|
| **NI_NS** | 100 mm | | | | |
| **NI_S** | 93 mm | **NI_S – NI_NS** | | -7 mm | -7 % |
| **IWU_S** | 87 mm | **IWU_S – NI_S** | | -6 mm | -7 % |
| **IAF_S** | 82 mm | **IAF_S – NI_S** | | -11 mm | -12 % |

We will integrate the findings in L246-L248 as follows:

*"Dralle et al. (2021) estimated that integrating snow accumulation and melt in the memory method led to an average reduction in $S_r$ of 6 mm (2%) for areas with >10% winter snow coverage, and 28 mm (17%) for areas with >80% winter snow coverage (Dralle et al., 2020b). These magnitudes are broadly consistent with our findings for irrigation (Table 2).* **Our results indicate that the effects of snow and irrigation on $S_r$ are comparable. In our study, 27 catchments have both considerable snowfall (snow days > 5% of the total days) and irrigation ($I_a$>0.05). For these catchments, the snow model (Appendix A) led on average to an $S_r$ reduction of 7 mm (7%) for the NI case compared to a set-up without the snow model. With irrigation, $S_r$ further reduced by 6 mm (7%) for IWU, and 11 mm (12%) for IAF in these catchments."**

*One of the key assumption of the methodology is a sustainable water use, but this is not the case in many locations, as mentioned L264-266. Beyond this sentence, a more detailed discussion of potential impact on the methodology (e.g. irrigation exceeding sustainable use by XX% implies XX% changes in S_r) would be quite interesting to put results into perspective*

We acknowledge that the assumption of sustainable water use is a limitation of the methodology used, as stated in L264-266. We expect that the potential impact of unsustainable water use on the methodology would be larger than the effects we found based on closing water balances. If water is unsustainably used for irrigation, more irrigation would be applied than according to our method, and, as a consequence, a larger reduction of $S_r$. Therefore, our estimates represent a lower limit of effects on $S_r$.

We will elaborate more in L261-266:

*"A further limitation may arise from the assumption in the memory method with irrigation methods proposed here that catchments are hydrologically closed systems. However, inter-catchment lateral flows, such as groundwater and irrigation water can significantly alter catchment water balances (e.g., Bouaziz et al., 2018; Fan, 2019; Condon et al., 2020). Moreover, the extraction of fossil groundwater for irrigation (Siebert et al., 2010; 265 Grogan et al., 2017; de Graaf et al., 2019) can violate the assumption of closing water balances for the here developed irrigation methods in the memory method.* **Our methodology based on a sustainable water use assumption provides a lower boundary of reduction in irrigated catchments. It is expected that irrigation exceeding sustainable use would lead to larger $S_r$ reductions than reported here, as in this case more water is available to crops than derived from the water balance."**

*Another assumption in the methodology is that there is single succession of excess/deficit periods within a year. How robust is that, also in relation to cited efforts to quantify root zone storage? How do other patterns such as double cropping system (where irrigation may happen in two periods), and hydroclimatic patterns such has bimodal monsoon (which e.g. affects significant parts of India) alter this framework?*

The memory method is based on the annual maximum storage deficits, and the irrigation model proposed here only focuses on the largest deficit each year. As suggested by the reviewer, patterns such as double cropping systems and bimodal monsoons are not directly accounted for in the method. We will integrate this limitation in L266:

"… irrigation methods in the memory method. **Furthermore, the methodology assumes single succession of excess and deficit periods within a year (Fig. 2b), which is not necessarily representative in regions with double cropping systems or bimodal monsoons (Biradar and Xiao, 2011)."**

*Between the ongoing irrigation expansion and improved irrigation efficiency, what would be the net effect on irrigation volumes and thus root zone storage estimates? Perhaps a more detailed perspective relating to McDermid et al. (2023) and other review literature on irrigation would be interesting*

We showed that the effect of irrigation on root zone storage estimates increases with increased irrigation water use (Fig. 7). Therefore, irrigation expansion, and thus increased irrigation water use, would lead to further reductions of $S_r$ at catchment scales in the near-future compared to the estimates reported in this study. On the other hand, the effect of improved irrigation efficiency is less straightforward. Improved irrigation efficiency (i.e., reduced soil evaporation) reduces the irrigation water volumes needed, which, at the catchment scale, leads to increased long-term mean discharge, and thus reduced long-term mean evaporation which leads to reduced $S_r$ in the memory method compared to a case with lower irrigation efficiency. However, it has been shown that increased efficiency does not necessarily lead to reduced irrigation water use, as the saved water by increasing irrigation efficiency is applied elsewhere (Grafton et al., 2018, Lankford et al., 2020). Therefore, the net effect of ongoing irrigation expansion and improved irrigation efficiency is not evident.

We will add the following lines in section 4.1 (L242):

**"Given the ongoing irrigation expansion as presented by McDermid et al. (2023), it is expected that larger irrigation water volumes lead to further reductions of $S_r$ at catchment scales in the near-future compared to the reductions reported in this study. At the same time, irrigation efficiency is also improving (McDermid et al., 2023), but this effect on $S_r$ is less straightforward. Improved irrigation efficiency (i.e., reduced soil evaporation) reduces the irrigation water volumes needed, which, at the catchment scale, leads to increased long-term mean discharge, and thus reduced long-term mean evaporation. This would result in reduced $S_r$ in the memory method compared to a situation with lower irrigation efficiency. However, it has been shown that increased efficiency does not necessarily lead to reduced irrigation water use, as the saved water by increasing irrigation efficiency is often applied elsewhere (Grafton et al., 2018, Lankford et al., 2020)."**

*Comment 1.5*

**L23-24:** *Vegetation also mediates soil evaporation and perhaps more importantly evaporated interception; these fluxes are generally smaller, but amount land evaporation to transpiration is misleading.*

We agree that we are missing the other evaporation fluxes than transpiration here. Therefore, we will modify L22-24 as follows:

*"Vegetation strongly influences the water cycle as it controls the partitioning of precipitation into **discharge and evaporation by mediating soil evaporation, interception evaporation and transpiration (Milly, 1994). Transpiration is defined as the water transport from the subsurface back to the atmosphere via the roots of vegetation,** and is, on average, the largest terrestrial water flux globally (Schlesinger and Jasechko, 2014)."*

*Comment 1.6*

**L41-42:** *Kleidon and Heimann (1998) and Kuppel et al. (2017) also used a similar approach, albeit using potential evaporation*

We acknowledge that both these references are strongly related to the memory method approach we used here. However, L41-42 describe very specific previous studies using the memory method based on actual evaporation, and we believe that the references based on potential evaporation do not fit here, but rather in the previous paragraph. In this paragraph, Kleidon and Heimann (1998) was already mentioned twice, and we will also integrate Kuppel et al. (2017) in this paragraph (L32).

*Comment 1.7*

**L160-164 / Fig. 3:** *since the RMSE(f_IAF, f_IWU) is computed for all catchments (for each beta value), why not showing the spread of RMSE (e.g. interquartile range) instead of a single line? Can you justify with beta should be constant for all catchments ? i.e., why not computing 4511 catchment-specific beta values that minimize the RMSE, as I doubt it is strictly 0.9 everywhere? Could the authors clarify this point, as the potential impact of this variability (or computing choice) upon S_r estimates in the IAF case could be quite interesting to discuss.*

In the IAF method we use a constant value for beta, that is derived from the irrigation water use data in combination with the irrigated area data. Beta was chosen as a constant to create a relatively simple approach that does not directly rely on the irrigation water use data, which is beneficial for application in time periods without irrigation water use data (both historical and future), as well as for regions where no reliable irrigation water use data is available. Moreover, using 4511 catchment-specific beta-values would in principle be the same as the IWU method, as beta is derived from the irrigation water use data. Figure 3 shows the RMSE for each beta value, and the line is a combination of many points. For each beta we have 4511 deviations between $f_{IAF}$ and $f_{IWU}$, and combining these deviations gives the RMSE for that specific beta. To clarify this approach, and to show the spread of the results, we will add another panel to the figure with the catchment specific $f_{IAF}$ and $f_{IWU}$ values for the optimal beta value of 0.9 (see Fig. C1).

To clarify the choice of a constant beta, we will modify L160-164 as follows:

*"with $I_a$ (-) the catchment irrigated area fraction and β (-) a correction factor that is constant in space and time for all catchments. **β was chosen as a constant to create a relatively simple approach that***

**does not directly rely on irrigation water use data, which is beneficial for application in time periods (both historical and future), without irrigation water use data, as well as for regions where no reliable irrigation water use data is available.** *We estimated $\beta$ by minimizing the difference between $f_{IAF}$ and $f_{IWU}$ in terms of Root Mean Squared Error (RMSE). We generated 1000 linearly spaced values for $\beta$ between 0 and 2.5, and computed $f_{IAF}$ for all the catchments. For all these cases, the RMSE of catchment $f_{IAF}$ and $f_{IWU}$ was computed (Fig. 3). The RMSE minimized for $\beta = 0.9$ (RMSE = 0.042), which is applied for all catchments in Eq. (6)."*

[Figure]

*Figure C1. (a) Root Mean Squared Error (RMSE) between the catchment irrigation factors fIWU (Eq. 5) and fIAF (Eq. 6) for 4511 catchments for 1000 linearly spaced values of $\beta$ between 0 and 2.5. $\beta_{opt}$ represents the value for $\beta$ where the RMSE minimizes. (b) Scatter of fIWU (Eq. 5) and fIAF (Eq. 6) for $\beta = \beta_{opt} = 0.9$ with lighter colours indicating a higher point density.*

**Comment 1.8**

*L169-179: I had to read this section several times to understand how S_r was finally derived, and I am not sure I did. It is announced in the first sentence but referring to a Table 1 which is actually more a reminder list of notations than explaining S_r. In the last sentence it is said that it is the mean of three values, while S_r is separately computed for NI, IWU, and IAF right? What is meant by "Sd,M-values with occurrences closest to T = 2 years"? Perhaps a supplementary figure with an example of S_r calculation across return periods for a (given set of) catchment(s) would help.*

We compute the $S_r$ separately for NI, IWU, and IAF. For each case, $S_r$ is computed as the average of the three annual maximum $S_d$ ($S_{d,M}$) values with occurrences closest to T=2 years (return levels of $S_{d,M}$ are presented by the crosses in Fig. 5b,d,f and h). To clarify the $S_r$ calculation we modified L176-179 as follows:

*"Here, we directly used the observed $S_{d,M}$ -values with occurrences closest to T=2 years instead of a fitted extreme value distribution, because fitting an extreme value distribution is ambiguous for return periods of interest (here: 2 years) much smaller than the timeseries length (here: >10 years). For all catchments, **for each irrigation case separately**, the $S_r$ was estimated as the mean of the three observed $S_{d,M}$ -values with occurrences closest to T=2 years, **as represented by the cross-markers closest to the vertical dashed line at T=2 years in Fig. 5b, d, f, and h."***

We clarified this issue in Fig. 5 (see Fig. C2 below) by changing the x-axis labels of (b), (d), (f) and (h).

[Figure]

*Figure C2. (a, c, e, g) Timeseries of storage deficits Sd (mm) (Eq. 1) for four illustrative catchments with increasing irrigation from top to bottom for the three irrigation cases NI, IWU, and IAF (Table 1) with for each catchment the associated annual mean irrigation water use (Iw), irrigated area fraction (Ia), and root zone storage capacity (Sr) values. (b, d, f, h) Return level plot of annual maximum storage deficits (Sd,M) (Eq. 7) for the three irrigation cases NI, IWU and IAF with the dashed vertical line corresponding to a return period T of 2 years (Section 2.2.1). The locations of the catchments are shown in Fig. 6. Catchment identity, continent, and Koppen Geiger climate zone are from top to bottom: br_0002356, South America, temperate (Cfb); ca_0000689, North America, continental (Dfb); es_0000742, Europe, Mediterranean (Csa); in_0000252, Asia, tropical (Aw).*

**Comment 1.9**

*L174-176: Here or in the Discussion, a tentative/summary (and if possible physically-based) explanation for why a 2-year return periods fits best would be welcome.*

Previous studies have found that different vegetation types adapt to droughts with different return periods (e.g. Wang-Erlandsson et al., 2016). For example, forests adapt to longer return periods (~40 years) by root investment than crops that are harvested each year. As described in L278-280, we selected a 2-year return period to represent both the yearly harvesting of crops, and the other vegetation in the catchment as the catchments are in no case entirely covered by crops.

We will change L174-176 from:

*"For croplands, and thus irrigated land, a return period of 2 years was found to the be the most representative (Wang-Erlandsson et al., 2016)"*

to:

*"Wang-Erlandsson et al. (2016) found that for croplands, and thus irrigated land, the best evaporation simulations with a global hydrological model were achieved with an $S_r$ based on a return period of 2 years, as croplands adapt to survive droughts with relatively short return periods."*

*Comment 1.10*

*L213-214: From this text it seems Fig. 8a only shows catchments with I_a > 0.05, but this is not mention in the Figure or its caption (contrary to Fig. S3b), could the authors clarify? I actually wonder if this selection if I_a > 0.05 does not also apply to parts of Table 2?*

Figure 8 shows indeed only catchments with $I_a$>0.05, which is mentioned in the caption:

*"…, and in (b) catchments are stratified based on climate zone (Sect. 2.2.2, Fig. S3), with for both (a) and (b) only catchments with irrigated area fraction $I_a$>0.05…."*

Also Table 2 represents catchments with $I_a$>0.05 for the region and climate part (not for the upper four rows). This was not clear in the caption, so we modified the caption of Table 2 as follows:

*"Median and interquartile range (IQR) of the relative $S_r$ difference ($\Delta_r S_r$ (%)) between the irrigation cases (IWU and IAF) and the no irrigation case (NI) with the catchments stratified for the top four rows based on irrigated area fraction ($I_a$) (Fig. 7), for the middle four rows based on region (**only catchments with $I_a$>0.05**) (Fig. 8a), and for the bottom five rows based on climate zones (**only catchments with $I_a$>0.05**) (Fig. 8b). IQR is given as the 25th percentile - 75th percentile."*

*Comment 1.11*

*L224-232: For this discussion, consider adding a third panel to Fig. 6 with the relative difference between IWU and IAF, to look for patterns?*

For completeness of the results, we will include the relative differences between IWU and IAF in the supplementary information of the manuscript (See Fig. C3), and refer to it in the text.

We will modify L224-225 as follows:

*"Figure 6 shows similar spatial patterns of $\Delta_r S_r$ for IWU and IAF, but the magnitudes differed. For most groups of catchments, IAF had a more pronounced effect on Sr than IWU (Table 2, **Fig. S4**)."*

[Figure]

*Figure C3. Relative difference in $S_r$ ($\Delta_r S_r$ (-)) for IWU compared to IAF ((IWU-IAF)/IAF). See Table 1 for details on the irrigation cases.*

*Comment 1.12*

*L268-270:* *Not necessarily true if irrigated area fractions increased since then.*

The study by McDermid et al. (2023) shows that the area equipped for irrigation considerably increased between 1960 and 2000. The catchments studied here have varying time periods between 1981 and 2010. The irrigation data we used is relatively 'late' in the 1981-2010 period (2011-2018 for $\overline{I_w}$ and 2005 for $I_a$). Therefore, the actual irrigation during the catchment time period was probably smaller than our estimates based on $\overline{I_w}$ and $I_a$. To clarify, we will modify L268-270 as follows:

*"The annual mean $\overline{I_w}$ used in IWU was based on the 2011-2018 period, while the catchment time series varied between 1981 and 2010. Similarly, the $I_a$ we used represented the 2005 irrigated area fraction (Siebert et al., 2015). The temporal* **mismatch between catchment hydrological timeseries and irrigation data** *may have led to an overestimation of I for the* **catchment** *specific period, as irrigated area, and irrigation techniques and efficiency have developed over the evaluated period (McDermid et al., 2023)."*

*Comment 1.13*

*L270-271:* *How did the authors conclude on the lack of impact? What does that refer to? At present, it is not very convincing and may perhaps be extended as part of the discussion (see General Comments).*

We acknowledge that the temporal mismatch between the data does impact the results. However, this research aimed to quantify how much irrigation in general would affect $S_r$, and to see patterns herein, which is not much influenced by the temporal mismatch of the data.

To clarify, we will modify L270-271 as follows:

*"The temporal* **mismatch between catchment hydrological timeseries and irrigation data** *may have led to an overestimation of I for the* **catchment** *specific period, as irrigated area, and irrigation techniques and efficiency have developed over the evaluated period (McDermid et al., 2023).* **Although this inconsistency in the temporal data influences the catchment specific outcomes, we believe that it** *did not have major influence on the quantification of the general patterns of the effects of irrigation on $S_r$, which was the aim of this study."*

**Technical comments**

*Comment 1.14*

*L185:* *Fig. S2-S3 are referred to before Fig. S1 (which is a referred though Fig. 4 a bit later). Perhaps consider swapping Fig. S1 for Fig. S3, etc. ?*

We will change the order of the supplementary figures as suggested.

*References*

*Kleidon, A., Heimann, M. (1998). A method of determining rooting depth from a terrestrial biosphere model and its impacts on the global water and carbon cycle. Glob. Change Biol. 4, 275–286. http://dx.doi.org/10.1046/j.1365-2486.1998.*

*Kuppel, S., Fan, Y., & Jobbágy, E. G. (2017). Seasonal hydrologic buffer on continents: Patterns, drivers and ecological benefits. Advances in Water Resources, 102, 178-187.*

*McDermid, S., Nocco, M., Lawston-Parker, P. et al. Irrigation in the Earth system. Nat Rev Earth Environ* **4**, *435–453 (2023).* https://doi.org/10.1038/s43017-023-00438-5

**References**

Biradar, C. M. and Xiao, X.: Quantifying the area and spatial distribution of double- and triple-cropping croplands in India with multi- temporal MODIS imagery in 2005, International Journal of Remote Sensing, 32, 367–386, https://doi.org/10.1080/01431160903464179, 2011.

Grafton, R. Q., Williams, J., Perry, C. J., Molle, F., Ringler, C., Steduto, P., Udall, B., Wheeler, S. A., Wang, Y., Garrick, D., and Allen, R. G.: The paradox of irrigation efficiency, Science, 361, 748–750, https://doi.org/10.1126/science.aat9314, 2018.

Kleidon, A. and Heimann, M.: A method of determining rooting depth from a terrestrial biosphere model and its impacts on the global water and carbon cycle, Global Change Biology, 4, 275–286, https://doi.org/10.1046/j.1365-2486.1998.00152.x, 1998.

Kuppel, S., Fan, Y., and Jobbágy, E. G.: Seasonal hydrologic buffer on continents: Patterns, drivers and ecological benefits, Advances in Water Resources, 102, 178–187, https://doi.org/10.1016/j.advwatres.2017.01.004, 2017.

Lankford, B., Closas, A., Dalton, J., López Gunn, E., Hess, T., Knox, J. W., van der Kooij, S., Lautze, J., Molden, D., Orr, S., Pittock, J., Richter, B., Riddell, P. J., Scott, C. A., philippe Venot, J., Vos, J., and Zwarteveen, M.: A scale-based framework to understand the promises, pitfalls and paradoxes of irrigation efficiency to meet major water challenges, Global Environmental Change, 65, 102 182, https://doi.org/https://doi.org/10.1016/j.gloenvcha.2020.102182, 2020.

McDermid, S., Nocco, M., Lawston-Parker, P., Keune, J., Pokhrel, Y., Jain, M., Jägermeyr, J., Brocca, L., Massari, C., Jones, A. D., Vahmani, P., Thiery, W., Yao, Y., Bell, A., Chen, L., Dorigo, W., Hanasaki, N., Jasechko, S., Lo, M.-H., Mahmood, R., Mishra, V., Mueller, N. D., Niyogi, D., Rabin, S. S., Sloat, L., Wada, Y., Zappa, L., Chen, F., Cook, B. I., Kim, H., Lombardozzi, D., Polcher, J., Ryu, D., Santanello, J., Satoh, Y., Seneviratne, S., Singh, D., and Yokohata, T.: Irrigation in the Earth system, Nature Reviews Earth & Environment, https://doi.org/10.1038/s43017-023-00438-5, 2023.

---

## Author Comment (AC2)

**Reviewer 2**

*The manuscript "Influence of irrigation on root zone storage capacity estimation" assesses the impact of global irrigation practices on root zone water storage capacity. The findings are quite interesting suggesting a general reduction in storage capacity particularly for agriculturally areas.*

*The paper is generally well written and fairly easy to understand given the theoretical nature and complexity of the topic. I find it suitable for publication in HESS after addressing some concerns.*

We would like to thank the referee for the comments. We appreciate the time and effort taken to read our manuscript in detail and to provide us the very useful and interesting thoughts on our research. We will take the comments into account when revising the manuscript.

We have separated the different comments (shown in *italic*) and have written our replies below. Text in the original manuscript is shown in '*italic*' and revised text in '**bold**'. Unless differently stated, line numbers mentioned in our reply refer to the original manuscript version.

**Comment 2.1**

*My main struggle when reading the manuscript was the lack of potential consequence of their estimations. For example, what are the consequences for landscape scale land-use and land management? I.e. you determined a decrease in root water storage capacity with irrigation, but would it not be more meaningful to try to explore "best" irrigation practices for a hydrologically resilient agriculture? I would prefer some calculations, but at least this issue should be thoroughly discussed.*

From an agricultural perspective it is indeed logical to explore the optimal irrigation practices in a way the crops can optimally function. However, we believe that the memory method as presented here is suitable for large-scales, such as catchments, but it may be too simplistic for the scale of agricultural fields. The aim of this study was to quantify the influence of irrigation on the root zone storage capacity at catchment scales, while wider implications for landscape scale land-use and land management are beyond the scope of this research.

**Comment 2.2**

*Discussion: in general the discussion is fairly short and not exactly spiked with literature comparison and contextualization. This could be improved. Aside from the suggestion above, one discussion point could be the process-based mechanisms underlying reduction in root water storage capacity. In the introduction the authors relate this mainly to anatomical changes in the rooting system, i.e. shallow and less dense root system under irrigation. However, plants react to changes in water input regime in more ways than anatomical adjustments. E.g. how does changes in hydraulics or generally differences in hydraulics between species affect Sr? Is it, i.e., possible that adjustments or species specific differences in plant maximum water potentials (ψ) affect Sr and how?*

We agree that we focused only on vegetation root responses to irrigation, without mentioning other plant adjustments. Plants react to irrigation activities also by changes in, for example, stomatal aperture (Chaves et al., 2016) or root hydraulic conduction (Gullo et al., 1998). In the discussion, we will also elaborate more on the process-based mechanisms underlying the here found reduction of root zone storage capacity as a result of irrigation. We will do this by adding the following lines after L239:

*"...of vegetation transpiration (Fig. 7).* **The reduction in $S_T$ in catchments with irrigation was expected following that the memory method is based on the theory that vegetation will invest less in roots if sufficient water is available (Guswa et al., 2008). The observed changes in $S_T$ are here attributed to**

*changes in the vegetation roots, as they are directly related to the size of $S_r$. Additionally, adaptations at the plant scale associated with irrigation, such as adjustments in stomatal aperture (Chaves et al., 2016) and root hydraulic conductance (Gullo et al., 1998), are also implicitly related to changes in $S_r$."*

**Specific comments**

*Comment 2.3*

*LL25: actually phenological development especially in croplands is pretty important and can easily outrule other influences.*

*LL25ff: this definition of Sr is dominated by physical objectives and does not consider plant regulation at all, same goes for the description of T (transpiration) regulation. This lacks an understanding of physiological and ecological processes that regulate T (transpiration) and I find this troublesome.*

We completely agree that phenological development also plays an important role in vegetation transpiration, though mostly at individual plant level. However, in this study we focus on entire ecosystems with mixed vegetation species, and approach the catchment vegetation transpiration from a large-scale water demand and supply perspective.

The definition of $S_r$ as the 'maximum volume per unit square of subsurface moisture that is accessible to roots of vegetation for uptake' is indeed mostly based on a physical objective, namely vegetation water supply at catchment scales. With respect to vegetation adaptivity, it is important to distinguish between individual plants, and the collective of individual plants within an ecosystem. Individual plants respond to droughts through for example root biomass adjustments, anatomical alterations, and physiological acclimations (e.g. Brunner et al., 2015). This adaptive capacity of individual plants depends on vegetation species (Zhang et al., 2020). Here, we focus on catchment scale where the root zone storage capacity represents the adaptation of the vegetation, i.e. the collective of all plants in the entire catchment with respect to subsurface water availability.

We will clarify this in the introduction in L25:

*"The amount and timing of vegetation transpiration **at catchment scales** is **largely** controlled by the interplay between seasonal energy and water availability signals (Gentine et al., 2012). **At individual plant scale, plants regulate transpiration also by root biomass adjustments, anatomical alterations, and physiological acclimation (e.g. Brunner et al., 2015), depending on vegetation species (Zhang et al., 2020). However, at the ecosystem scale, which represents the collective of individual plants,** the subsurface water removal by transpiration is regulated by the liquid water input and by the available subsurface water buffer. This water buffer, the root zone storage capacity ($S_r$), is defined as the maximum volume per unit square of subsurface moisture that is accessible to roots of vegetation for uptake (Gao et al., 2014)."*

*Comment 2.4*

*LL44: do you truly mean evaporation or evapotranspiration?*

We acknowledge that various perspectives exist concerning the definition of evaporation vs evapotranspiration. Here we mean evaporation, defined as the sum of transpiration, soil evaporation, and interception evaporation. We will clarify L44 as follows:

*"…seasonal signals of precipitation and evaporation, **here defined as the total of transpiration, soil evaporation, and interception evaporation, following the terminology proposed by Savenije (2004) and Miralles et al. (2020).**"*

*Fig 2 and methods section: Why do you specifically need two years? Also: You start the hydrological year with the day of highest water availability. But how do you deal with consecutive years varying in precipitation regime? Or do you just define this for the starting point?*

Figure 2 only shows two years of a timeseries to illustrate the method, but all catchments have at least ten years of data available. We will clarify this in the caption of Fig. 2 as follows:

*"…(b) An example time series of $S_s$, $S_d$ and I based on Eqs. (1-6) with $\Delta t_d$ the length of the deficit period (days), and $S_s$ (ts1) the surplus storage at the end of the surplus period. **Note that this time series represents only two years to illustrate the method, while all catchments have at least ten years of data.**"*

We start the hydrological year on the day of highest water availability, but this is only used as starting point on the first day of the full timeseries. This means that during consecutive years with varying precipitation regimes the storage deficits do not necessarily recover each year. We will clarify this in L120 as follows:

*"In Eq. (1) t0 corresponds to the first day of the first hydrological year and $\tau$ to the daily time steps ending on the last day of the last hydrological year. Our hydrological year starts the first day of the month after the wettest month, which is defined as the month with on average the largest positive difference between monthly mean P and Ep. **At t0, the starting point of the analysis, $S_d$=0.**"*

*Fig. 4 and 6: the way the figure is plotted in the preprint this is very hard to read given the size and color palette.*

For Fig. 4 we will change the colormap to the matplotlib 'cubehelix' colormap, which is colorblind-proof, and covers a relatively large lightness-range. See the updated figure in Fig. C1.

For Fig. 6, we believe that the here used colormap represents our intentions with the figure well, as we want to emphasize on the catchments where the $\Delta S_r$ is relatively large (the darker, the larger), while still showing the catchments with small $\Delta S_r$ (yellow).

[Figure]

*Figure C1. Catchment Sr for the No Irrigation (NI) case, with dots representing catchment outlets. Similar figures for the IWU and IAF cases are presented in Fig. S1.*

**References**

Brunner, I., Herzog, C., Dawes, M. A., Arend, M., and Sperisen, C.: How tree roots respond to drought, Frontiers in Plant Science, 6, https://doi.org/10.3389/fpls.2015.00547, 2015.

Chaves, M., Costa, J., Zarrouk, O., Pinheiro, C., Lopes, C., and Pereira, J.: Controlling stomatal aperture in semi-arid regions—The dilemma of saving water or being cool?, Plant Science, 251, 54–64, https://doi.org/10.1016/j.plantsci.2016.06.015, special Issue: Water-Use Efficiency in Plants, 2016.

Lo Gullo, M. A., Nardini, A., Salleo, S., and Tyree, M. T.: Changes in root hydraulic conductance (KR) of Olea oleaster seedlings following drought stress and irrigation, New Phytologist, 140, 25–31, https://doi.org/10.1046/j.1469-8137.1998.00258.x, 1998.

Savenije, H. H.: The importance of interception and why we should delete the term evapotranspiration from our vocabulary, Hydrological processes, 18, 1507–1511, https://doi.org/10.1002/hyp.5563, 2004.

Miralles, D. G., Brutsaert, W., Dolman, A. J., and Gash, J. H.: On the Use of the Term "Evapotranspiration", Water Resources Research, 56, e2020WR028 055, https://doi.org/10.1029/2020WR028055, 2020.

Zhang, B., Hautier, Y., Tan, X., You, C., Cadotte, M. W., Chu, C., Jiang, L., Sui, X., Ren, T., Han, X., and Chen, S.: Species responses to changing precipitation depend on trait plasticity rather than trait means and intraspecific variation, Functional Ecology, 34, 2622–2633, https://doi.org/https://doi.org/10.1111/1365-2435.13675, 2020.